# High-Precision Measurements of the Bound Electron's Magnetic Moment

**Sven Sturm [1,\*], Manuel Vogel [2], Florian Köhler-Langes [1], Wolfgang Quint [2], Klaus Blaum [1] and Günter Werth [3]**

[1] Max Planck-Institut für Kernphysik, 69117 Heidelberg, Germany; florian.koehler@mpi-hd.mpg.de (F.K.-L.); klaus.blaum@mpi-hd.mpg.de (K.B.)
[2] GSI Helmholtzzentrum für Schwerionenforschung, 64291 Darmstadt, Germany; M.Vogel@gsi.de (M.V.); W.Quint@gsi.de (W.Q.)
[3] Johannes Gutenberg-Universität, 55099 Mainz, Germany; werth@uni-mainz.de
\* Correspondence: sven.sturm@mpi-hd.mpg.de; Tel.: +49-6221-516-447

**Abstract:** Highly charged ions represent environments that allow to study precisely one or more bound electrons subjected to unsurpassed electromagnetic fields. Under such conditions, the magnetic moment (*g*-factor) of a bound electron changes significantly, to a large extent due to contributions from quantum electrodynamics. We present three Penning-trap experiments, which allow to measure magnetic moments with ppb precision and better, serving as stringent tests of corresponding calculations, and also yielding access to fundamental quantities like the fine structure constant *α* and the atomic mass of the electron. Additionally, the bound electrons can be used as sensitive probes for properties of the ionic nuclei. We summarize the measurements performed so far, discuss their significance, and give a detailed account of the experimental setups, procedures and the foreseen measurements.

**Keywords:** electron magnetic moment; highly charged ion; quantum electrodynamics

## 1. Introduction

The magnetic moment of the electron is a fruitful object of high-precision studies due to the existence of both highly precise experiments and of comparatively precise predictions in the framework of quantum electrodynamics (QED). The magnetic moment is often expressed by the dimensionless "*g*-factor", which measures the magnetic moment in units of the Bohr magneton. In Dirac theory, a point-like free electron has exactly $g = 2$ [1]. Experiments by Kusch and Foley first found a deviation of *g* from 2 by measuring $g = 2(1.00119 \pm 0.00005)$ [2,3], which could be explained by Schwinger [4] when considering the self-energy effect of quantum electrodynamics. In respect to this deviation from 2, one often speaks about the "anomaly" $a = (g-2)/2$ of the magnetic moment. A number of experiments measuring magnetic moments have subsequently been performed (for a historic overview, see [5,6]). Particularly noteworthy is the measurement of the electron and positron anomalies carried out by Van Dyck et al. [7], which have yielded the values $a_e = 0.001\,159\,652\,188\,4\,(43)$ (for the electron) and $a_p = 0.001\,159\,652\,187\,9\,(43)$ (for the positron). Assuming exact CPT invariance, the two values can be combined to give an electron value of $g = 2.002\,319\,304\,376\,6\,(84)$, or $a = 0.001\,159\,652\,188\,3\,(42)$. For nearly twenty years, this value was the most precise available until the studies of Gabrielse et al., leading to a value of $g = 2.002\,319\,304\,361\,460\,(56)$, or $a = 0.001\,159\,652\,180\,73\,(28)$ [8]. QED theory very tightly relates this number to the value of the fine structure constant *α* [9], which has resulted in the most precise value of $\alpha^{-1} = 137.035\,999\,084\,(51)$ so far [8], in agreement with the best determination of $\alpha^{-1} = 137.035\,999\,037\,(91)$ employing an unrelated technique based on the measurement of the

recoil momentum of photons that are absorbed by atoms [10]. These findings are commonly seen as the most convincing corroborations of the validity of QED theory in the free-particle case, i.e. in the absence of external fields.

In the presence of external fields, for example when the electron is bound to a nucleus in the case for ions, there are significant changes to the value of its magnetic moment. Apart from relativistic effects as described analytically for the electron bound in a hydrogen-like ion by Breit [11], the most significant contributions come from QED and nuclear effects. This is not surprising since the field strengths at the location of the electron reach up to about $10^{16}$ V/cm (close to the Schwinger limit) and $10^5$ T for high values of the nuclear charge number $Z$ [12]. In addition, due to the tight binding of the electron to the nucleus, it becomes sensitive to nuclear properties [13,14]. Calculations of bound-electron magnetic moments [12–23] have been complemented by measurements [24–29], which will be described in detail below. Precise measurements of electron magnetic moments in highly charged ions allow for inferring nuclear properties in a way otherwise impossible due to the absence of diamagnetic shielding when only one or few electrons are present [30]. They allow for determining the atomic mass of the electron with unprecedented accuracy [31,32], yield an alternative access to the fine structure constant [33,34], and represent the most stringent tests of calculations in the framework of strong-field quantum electrodynamics [35].

In the following, we present three experiments concerned with corresponding measurements. They complement one another by the range of ion charge states available and by the choice of the measurement method. In combination, they allow access to a broad range of ion species and charge states. One is the 'Mainz $g$-factor experiment', which has successfully been used to perform measurements with $^{12}C^{5+}$ [24], $^{16}O^{7+}$ [25], $^{28}Si^{13+}$ [26], $^{28}Si^{11+}$ [27] and $^{40,48}Ca^{17+}$ [29]. It employs the so-called 'continuous Stern–Gerlach effect' [36,37], which is the method of choice for ions with zero-spin nuclei. The highly charged ions are created inside the Penning-trap setup used for the measurement, which sets an upper limit on the available charge states. An extension of this experiment, named ALPHATRAP, is currently being set up at MPI-K Heidelberg. It uses similar measurement techniques but receives ions from an external electron beam ion trap (EBIT) that can produce much higher charge states. The third experiment, currently under commissioning, is ARTEMIS [38,39], located at the HITRAP [40] facility at GSI, Darmstadt. It employs a spectroscopic technique for ions with non-zero nuclear spin and is designed to work with ions up to the highest charge states.

## 2. Sources of Highly Charged Ions

The kind of ion source used for experiments on few-electron ions depends mainly on the nuclear charge and the ionic charge state of interest.

Few-electron ions of low- and medium-Z elements are commonly created by electron-impact ionization in EBIT-type (Electron Beam Ion Trap) sources that are either part of or closely attached to the experimental setup that is used to perform the $g$-factor measurements. In contrast, in the high-Z region, such ions have to be produced in a source outside and then injected into the apparatus. In the $g$-factor experiments on highly charged ions performed so far in Mainz, atoms and singly charged ions are released from a surface covered by the element of interest inside the trap setup by an impinging electron beam. Consecutive ionisation of these ions by the same electron beam leads to the requested charge state. This so-called 'creation trap' is part of the sequence of Penning-traps as central instruments of the $g$-factor setup. Electron beams of a few hundreds of nA and energies up to a few keV are sufficient to create the desired charge state within a few seconds. The obvious advantage is that the ion creation device is placed in the same vacuum container as the measuring device. It is, however, limited to electron energies of about 5 keV in order to prevent breakdown of electrical insulation. The highest charge state so far obtained are 13+ in case of silicon (ionization potential 2.4 keV) and 17+ for calcium (ionization potential 1.1 keV).

For few-electron medium-Z ions, a conventional EBIT (Electron Beam Ion Trap) may be employed [41]. Here, the charge states are produced by consecutive ionisation of ions confined

mainly in a high-intensity electron beam of several keV energy in the presence of a magnetic field. EBITs are also used as sources by extracting the ions from them after breeding [42]. Compact EBIT/S using permanent magnets and electron energies up to 15 keV ('Dresden EBIT') are commercially available. They produce charge states up to 46+ in the case of xenon. They can be attached to the experimental setup, and ions can be injected into the measurement region. The vacuum inside such an EBIT, however, does not match the requirements for the *g*-factor experiments. Consequently, a valve has to separate the EBIT from the remaining parts, and has to be actuated fast in order to prevent too many background molecules from entering the measurement part during the injection time. The fact that the valve has to operate at cryogenic temperatures also represents a technical challenge.

For even higher charge states, the ionising electron beam of the EBIT needs higher energies than can be obtained by a compact or commercial design. Separated large-scale machines have to be used to produce those ions, which then need to be injected into the *g*-factor experiment. One such EBIT is operated at the Max Planck Institute for Nuclear Physics in Heidelberg, Germany. Thus far, it has been demonstrated to produce highly charged ions up to $Hg^{78+}$ [43]. Another possibility for creation of ions up to the highest charge states, particularly when larger ion quantities are desired, is accelerator-based electron stripping and subsequent slowing of the ions for dynamic capture into the trap. To this end, the HITRAP facility [40] at GSI, Germany, has been designed. Routinely produced ions of any species as available at GSI are to be decelerated in dedicated structures and cooled to 4 K in a separate Penning-trap. These ions can then be transported via a low-energy beamline to other Penning-trap experiments. This facility is currently undergoing commissioning.

## 3. *g*-Factor Measurements with Highly Charged Ions

### 3.1. Mainz g-Factor Experiment

Experiments have been performed on single hydrogen-like ions $^{12}C^{5+}$, $^{16}O^{7+}$, and $^{28}Si^{13+}$ as well as on lithium-like ions $^{28}Si^{11+}$, $^{40}Ca^{17+}$ and $^{48}Ca^{17+}$. In either case, one ion is confined in a cylindrical Penning-trap (Figure 1) with a superimposed homogeneous magnetic field of 3.76 T. We use a five-pole trap with a central ring electrode of 7 mm inner diameter, two endcaps, and two correction electrodes placed between them. By choosing the proper voltages at these electrodes, the trapping potential can be made highly harmonic near the trap centre. A confined ion with small motional amplitudes performs harmonic oscillations about the trap centre in the radial and axial directions, the frequencies of which depend on the trap's operating conditions. Typical values are 700 kHz for the axial oscillation frequency $\omega_z/2\pi$. In the radial plane, we have a so-called 'perturbed cyclotron frequency' $\omega_+/2\pi$ of typically 25 MHz and a slow 'magnetron frequency' $\omega_-/2\pi$ of a few kHz.

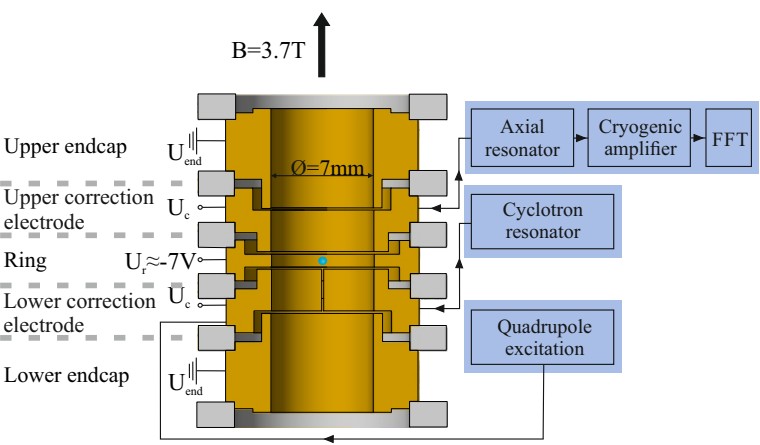

**Figure 1.** Five-electrode cylindrical Penning-trap with attached electronics for ion detection and cooling.

The trap is located at the centre of a superconducting solenoid of high homogeneity and temporal stability. To avoid charge exchange of the trapped ion due to a collision with residual gas, the trap is cooled to 4 K, which assures efficient cryo-pumping. A pressure of better than $10^{-16}$ mbar in the trap volume has been demonstrated. Consequently, ion storage times of many months are routinely achieved. Attached to the trap electrodes are superconducting resonance circuits of high quality factors. They serve for trapped-ion detection by measuring the image currents induced in the trap's electrodes. Apart from the trap, the resonance circuits and the associated electronics are also held at 4 K to reduce thermal noise. Details of the experimental setup can be found in Reference [28]. Keeping the ion oscillation frequencies in resonance with the detection circuits leads to ion cooling by energy dissipation into the circuits ('resistive cooling'). A detection of the ion in thermal equilibrium with the environment is performed by a Fourier transform of the circuit's noise power: The oscillating ion can be represented as a series resonance circuit of very high quality in parallel to the detector. This circuit shortcuts the noise voltage of the detection circuit and leads to a minimum ('dip') in the noise spectrum. Figure 2 shows, as an example, the signal from the axial oscillation of a single $^{12}C^{5+}$ ion. The central part of the Fourier spectrum is shown, demonstrating the small width of the dip of about 500 mHz in a total oscillation frequency of $\omega_z/2\pi = 670$ kHz.

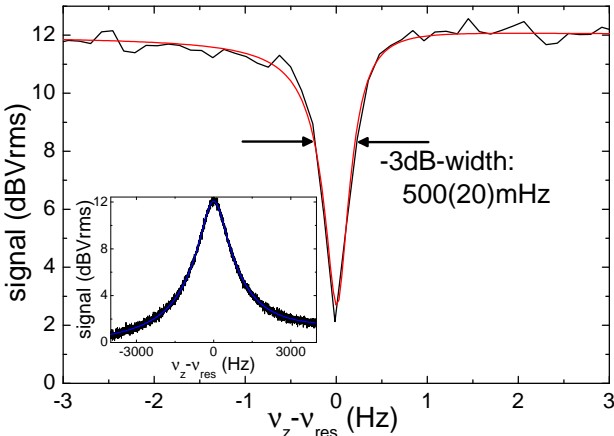

**Figure 2.** Fourier transform of the noise voltage across the axial resonance circuit attached to a trap electrode in the presence of a single trapped $^{12}C^{5+}$ ion. The ion shortens the noise of the resonator and generates a dip signal with a full width at half maximum (FWHM) of 0.5 Hz. The total oscillation frequency of about 670 kHz can be determined from the fitted lineshape (red line) with an uncertainty of 100 mHz. The insert shows the complete resonator and the corresponding fit (blue line).

The axial resonance serves also for the determination of the radial frequencies that are coupled to the axial one by an rf-field at their frequency difference to the axial oscillation. The knowledge of all three frequencies is required to calibrate the magnetic field B at the ion's position through the free-ion cyclotron frequency $\omega_c = qB/M$, where $q$ is the ion's charge and $M$ is the ion's mass. $\omega_c$ is obtained via the relation $\omega_c^2 = \omega_+^2 + \omega_z^2 + \omega_-^2$ [44]. The bound electron's g-factor is determined by a measurement of the Larmor precession frequency

$$\omega_L = \frac{g}{2}\frac{e}{m}B,\tag{1}$$

where $e$ is the charge and $m$ is the mass of the electron. Transitions between the two spin states of the electron are induced by a microwave field at a frequency of (nominally) 104 GHz. In a perfect Penning-trap, the spin precession is completely decoupled from the ion's oscillation and cannot be measured directly. Hence, we induce coupling by making the trap's magnetic field inhomogeneous on account of a ferromagnetic ring electrode that locally distorts the otherwise homogeneous magnetic

field ("magnetic bottle"). To first order, it leads to a quadratic dependence of the *B*-field on the spatial coordinate: $B = B_0 + B_2 z^2 + \ldots$. The additional force $F = \mu_z \nabla_z B$ of the inhomogeneous *B*-field acting on the spin magnetic moment adds to or subtracts from the electric trapping force, depending on the spin direction, and leads to a change in the oscillation frequencies ('continuous Stern–Gerlach effect' [36]). For the axial direction, the frequency change is

$$\Delta \nu_z = \frac{g \mu_B}{(2\pi)^2 M \nu_z} B_2, \tag{2}$$

where $\mu_B$ is the Bohr magneton and $B_2 = 10,500\,\text{T}/\text{m}^2$ is the strength of the magnetic bottle. In the case of the electron bound in hydrogen-like carbon C$^{5+}$, it amounts to $\Delta \omega_z / 2\pi = 580\,\text{mHz}$ (Figure 3).

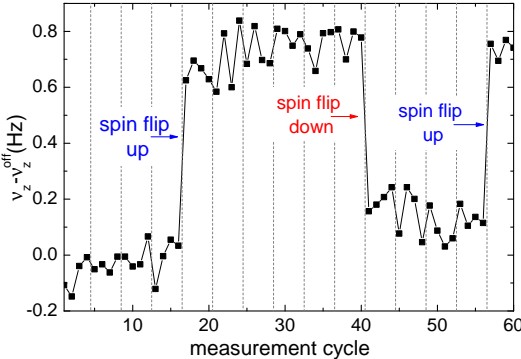

**Figure 3.** Detection of three induced spin flips of a single $^{12}$C$^{5+}$ ion by a change of the ion's axial oscillation frequency. The frequency change amounts to 0.58 Hz in a total frequency of 411 kHz. Every four axial frequency measurement cycle microwaves around the Larmor frequency are induced.

The introduction of the magnetic field inhomogeneity as required for detection of the spin direction prevents high accuracy in the determination of the oscillation frequencies as well as the Larmor precession frequency. To circumvent this problem, the region where the frequencies are measured ("precision trap") is separated from the region where the spin state is analyzed by adding a second trap ("analysis trap") of identical dimensions but without the ferromagnetic ring. The distance between the two trap centres is 3.5 cm. The setup is shown in Figure 4.

The ion can be shifted between the traps by moving the potential minimum along the axis by appropriate changes of the voltages at the electrodes. A measurement cycle starts with the determination of the spin direction in the analysis trap. The ion is then transported into the precision trap where the motional frequencies are determined, and, simultaneously, an attempt is made to induce electronic spin flips. After transport back to the analysis trap, the spin direction is probed to determine whether the attempt at a given microwave frequency was successful, and the procedure is repeated for different microwave frequencies. The hence measured probability for spin flip success as a function of the ratio $\Gamma = \omega_L / \omega_c$ has a maximum at $\Gamma_0$ from which the *g*-factor is determined through

$$g = 2\Gamma_0 \frac{q}{e} \frac{m}{M}. \tag{3}$$

The experimental results for $^{12}$C$^{5+}$ ($g = 2.001\,041\,592\,44\,(232)$), $^{16}$O$^{7+}$ ($g = 2.000\,047\,025\,4\,(46)$), and $^{28}$Si$^{13+}$ ($g = 1.995\,348\,959\,04\,(81)$) agree very well with the theoretical predictions (see Table 1). They represent to date the most stringent test of bound-state QED calculations.

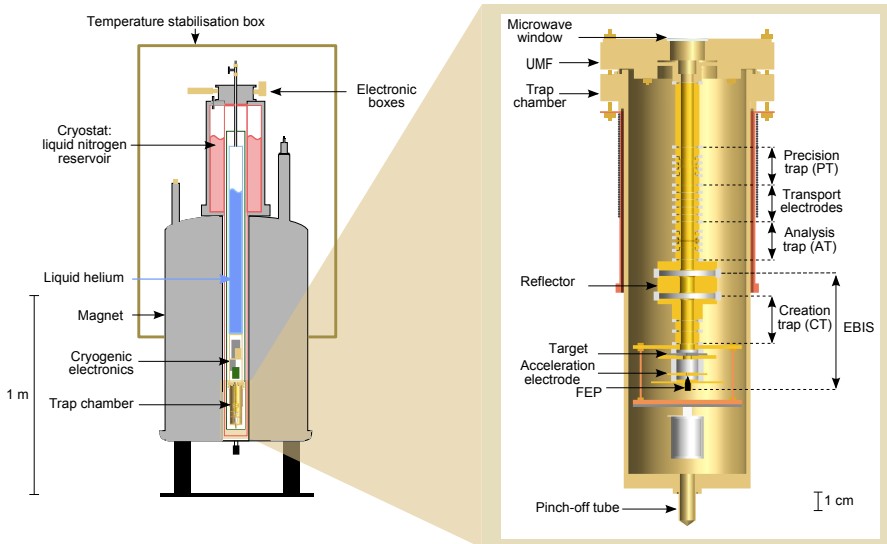

**Figure 4.** Illustration of the trap setup. (**Left**) Overview of the superconducting magnet setup with the Penning-trap located in the magnetic field center; (**Right**) Trap chamber with Penning-trap setup.

**Table 1.** Calculated contributions to the *g*-factor of the electron bound in different hydrogen-like ions [31] and comparison to experimental values. The uncertainty in the theory reflects the estimates on the size of uncalculated higher-order terms. For comparison, the experimental results are listed. The uncertainties of the experimental values include the uncertainty of the electron mass at the time of the measurement.

| Contribution | $^{12}\text{C}^{5+}$ | $^{16}\text{O}^{7+}$ | $^{28}\text{Si}^{13+}$ |
|---|---|---|---|
| Dirac value | +1.998 721 354 391 (1) | +1.997 726 003 06 (2) | +1.993 023 571 551(5) |
| free QED | +0.002 319 304 358 (1) | +0.002 319 304 358 (1) | +0.002 319 304 358 (1) |
| BS-QED | +0.000 000 843 391 (6) | +0.000 001 594 38 (11) | +0.000 005 855 67 (165) |
| nuclear size | +0.000 000 000 407 | +0.000 000 001 55 (1) | +0.000 000 020 468 |
| nuclear recoil | +0.000 000 087 629 | +0.000 000 116 97 | +0.000 000 205 881 |
| theory total | +2.001 041 590 117 (6) | +2.000 047 021 28 (11) | +1.995 348 957 93 (165) |
| experiment | +2.001 041 592 44 (232) | +2.000 047 025 4 (46) | +1.995 348 959 04 (81) |

In the experimental results as listed in Table 1, the largest contribution to the error budget arises from the uncertainty of the electron mass. The validity of the *g*-factor calculation has been tested on the level of $10^{-10}$ in the case of $^{28}\text{Si}^{13+}$ (see Table 1). The QED part scales approximately with the square of the nuclear charge *Z*. Consequently, they are much smaller for the case of $^{12}\text{C}^{5+}$ and its quoted theoretical uncertainty of $1.5 \times 10^{-11}$ can be considered as correct. Assuming the QED predictions to be exact, we can then rewrite Equation (3) as

$$m = \frac{g_{th}}{2} \frac{1}{\Gamma_0} \frac{e}{q} M \tag{4}$$

to determine the electron's atomic mass using the carbon's atomic mass *M* as reference. The result of $m = 0.000\,548\,579\,909\,069\,4\,(155)$ [31] improves the present value listed in the CODATA table of fundamental constants by a factor of 13.

The finite nuclear size contribution to the *g*-factor as listed in Table 1 depends on the charge radius of the corresponding nucleus. Considering the remaining contributions as correct and using the new value for the electron mass, we can extract a value for the nuclear radius $< r^2 >^{1/2}$ from a comparison of theoretical and experimental results. For the case of $^{28}\text{Si}^{13+}$, we obtain $< r^2 >^{1/2} = 3.18\,(15)$ fm in agreement with the value $< r^2 >^{1/2} = 3.18223\,(24)$ fm from nuclear scattering [45]. Although our value

is not competitive with the scattering result, it can be considered as proof-of principle to demonstrate the feasibility of this method for nuclear radius determination, which otherwise may not be obtainable.

The high precision obtained in the determination of the *g*-factor may require a more accurate theoretical calculation going beyond the standard formalism of bound-state QED. Most contributions to *g*-factors of highly charged ions, e.g., the relativistic, radiative, nuclear size or, in the case of lithium-like ions, interelectronic interaction corrections are calculated using BS-QED in the infinite-nuclear-mass approximation. Here, the nucleus is considered as an external Coulomb potential fixed in space. This approach is usually referred to as the Furry picture of QED [46]. The calculation of the studied nuclear recoil shift, however, requires going beyond the Furry picture [46]. As a first example, we investigated the isotope dependence of the Zeeman effect by studying *g*-factors of lithium-like calcium isotopes $^{40}$Ca$^{17+}$ and $^{48}$Ca$^{17+}$. These isotopes are particularly well-suited because they have almost identical nuclear charge radii [47] in spite of their 20% mass difference. Therefore, they provide a unique system across the entire nuclear chart to test the relativistic nuclear recoil shift. The experiments were carried out in the same manner as described above for H-like ions. The corresponding results are $g_{ex}(^{40}\text{Ca}^{17+}) = 1.999\,202\,040\,5\,(11)$ and $g_{ex}(^{48}\text{Ca}^{17+}) = 1.999\,202\,028\,85\,(82)$ in agreement with the corresponding theoretical values $g_{th}(^{40}\text{Ca}^{17+}) = 1.999\,202\,042\,(13)$ and $g_{th}(^{48}\text{Ca}^{17+}) = 1.999\,202\,03\,2\,(13)$, respectively, representing a test of the description of the inter-electronic interaction between the three electrons in the Li-like system [29]. The *g*-factor difference has been experimentally determined from the measured frequency ratios $\Gamma_0 = \omega_L/\omega_c$ to

$$\Delta g_{ex} = g_{ex}(^{40}\text{Ca}^{17+}) - g_{ex}(^{48}\text{Ca}^{17+}) = 11.70\,(16)\,(3)\,(138) \times 10^{-9}, \tag{5}$$

where the first error bars represent the statistical, the second one the systematical uncertainty, while the third one arises from the mass measurements of the calcium isotopes which dominate the total uncertainty. The comparison of the measured value of the *g*-factor difference with the theoretical prediction

$$\Delta g_{th} = g_{th}(^{40}\text{Ca}^{17+}) - g_{th}(^{48}\text{Ca}^{17+}) = 10.305\,(27) \times 10^{-9} \tag{6}$$

allows, for the first time, a direct test of the relativistic interaction of the electron spin with the motile nucleus in the presence of strong Coulomb fields and confirms the calculation at the 10% level.

### 3.2. ALPHATRAP

While the creation of highly charged ions up to hydrogen-like silicon $^{28}$Si$^{13+}$ is possible in situ in the Mainz setup, in order to test QED in the most extreme fields, it is desirable to access even heavier systems, up to hydrogen-like $^{208}$Pb$^{81+}$. To this end, the hermetically sealed trap design has to be abandoned in favour of a connection to a room-temperature vacuum in order to allow injection of arbitrary species from an external source. For this purpose, a novel Penning-trap setup, ALPHATRAP, has been developed at the Max Planck Institute for Nuclear Physics (MPIK) in Heidelberg, Germany. ALPHATRAP comprises a cryogenic double-trap system (see Figure 5), including a dedicated capture section. The trap section is connected to a UHV/XHV (ultra-/extremely high vacuum) beamline with electrostatic ion optics to allow the efficient transfer of ions from different sources to the Penning-traps. In order to limit the influx of rest-gas atoms from the room-temperature beamline, a vacuum valve has been developed which can be operated at cryogenic temperatures. This valve seals off the trap section with solely cryogenic surfaces.

Compared to the current Mainz *g*-factor trap, the trap diameter has been increased from 7 mm to 18 mm. This reduces the dominant systematic shift, which arises from the image charges in the electrodes, and which scales with the third power of the diameter, here by a factor of 17. Furthermore, the trap design, which features two sets of correction electrodes, has been extensively optimized for excellent electrostatic and magnetic field quality. Combined with the newly developed phase-sensitive

detection technique PnA [35], this paves the way for a measurement of the *g*-factor of the bound electron with a relative precision in excess of 10 parts per trillion.

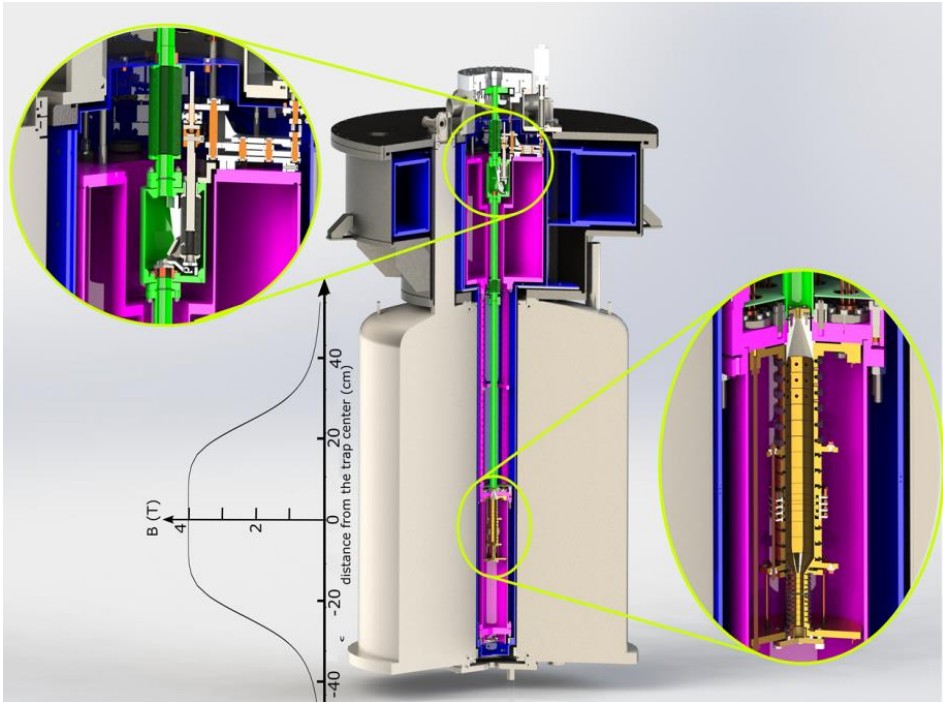

**Figure 5.** The ALPHATRAP Penning-trap setup. The trap chamber (**right inset**) is connected to the room-temperature beamline (green); A cryogenically operateable valve (**left inset**) allows for sealing the cryogenic section during the measurement. This limits the in-flux of rest-gas atoms from the room-temperature section and allows reaching the close-to-perfect vacuum conditions required for the storage of heavy highly-charged ions. To provide optimal mechanical stability, the cryogenic section is cooled by a specially developed liquid helium cryostat.

At the MPIK, the Heidelberg EBIT [43] is currently offering access to medium-heavy ions with ionisation energies in excess of 60 keV [48,49]. Combined with the newly developed electronics for single ion detection, ALPHATRAP provides access to all elements and charge states in the medium- to high-*Z* regime. The spin-state detection using the continuous Stern–Gerlach effect, as well as the eigenfrequency detection via the image current, are designed to work for all contemplable elements and charge states, which makes ALPHATRAP very flexible and enables access to a multitude of measureable quantities and contributions of bound-state QED. In particular, the Stern–Gerlach type detection works also with ions without nuclear spin, for which the precision of the theory is far superior. This way, ALPHATRAP is aiming to perform the most stringent test of QED in strong fields. Besides the test of QED theory, the dependence of *g* on the fine structure constant *α* opens the possibility for an alternative determination of this fundamental constant using the theoretical and experimental values. The main contribution to the *g*-factor of hydrogen-like ions arises from the solution of the Dirac equation

$$g_D = 2 - \frac{2}{3}(\alpha Z)^2 - \frac{1}{6}(\alpha Z)^4 + \dots \tag{7}$$

The uncertainty in *α* depends on the uncertainties of $g_{th}$ and $g_{ex}$ in the first order as

$$\frac{\delta \alpha}{\alpha} \approx \frac{3}{2} \frac{1}{(\alpha Z)^2} \sqrt{(\delta g_{th})^2 + (\delta g_{ex})^2}. \tag{8}$$

From Equation (8), a high-Z element looks most promising for an accurate determination of $\alpha$. However, a major limitation stems from nuclear size and structure effects. To a large extent, these contributions can be cancelled in a specific difference of $g$-factors of ions in different charge states but with the same nucleus, as proposed in [33]. This way, it seems possible to extract a value for $\alpha$ which is largely independent of the currently tabulated values. However, the achievable precision will not be sufficient for matching or surpassing the accuracy of the presently most accurate $\alpha$ determinations from the free electron's $g$-factor or from atomic recoil experiments [8,10]. Instead, it seems possible to use light ions, for which the theoretical prediction is substantially more precise. Since the sensitivity to $\alpha$ is significantly lower in the low-Z regime, the experimental precision will have to be improved far beyond the current state-of-the-art, well below the parts-per-trillion regime. While this precision is currently out of reach for the absolute value of the $g$-factor, a novel measurement scheme has been developed to allow the direct measurement of differences of $g$-factors:

$$\delta g_D(Z_1, Z_2) = \frac{2}{3}\alpha^2(Z_1 - Z_2)^2 + \dots \tag{9}$$

The quantity $\delta g_D$ is by far more sensitive to $\alpha$. With a foreseeable improvement in theory, this will enable a competitive and independent determination of $\alpha$, one of the most prominent fundamental constants of the Standard Model.

Currently, the ALPHATRAP experiment is being commissioned at the MPIK. To this end, several ion sources are connected to the trap. For the commissioning of the traps and the detection electronics, an in-trap EBIT similar to the design in the Mainz setup is implemented [50]. The injection from the room-temperature section into the trap can be optimized with a room-temperature permanent magnet EBIT [51]. From this EBIT ("tt-EBIT"), highly charged ions have been already successfully extracted and transported to the end of the beamline (see Figure 6). Once the trap has been commissioned, these ions are available to be re-trapped in the cryogenic trap. Finally, the connection of the room-temperature beamline to the Heidelberg EBIT ("HD-EBIT") is currently being developed, which will provide access to the full spectrum of highly charged ions available from that source.

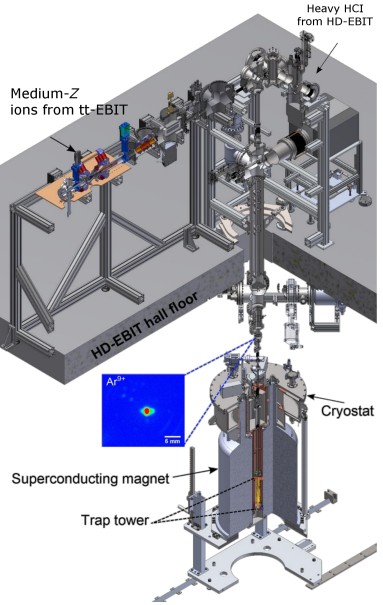

**Figure 6.** Overview of the ALPHATRAP ion sources. The tt-EBIT provides few-electron ions of low to medium Z. Recently, the extraction and transport of highly charged ions from the tt-EBIT to the end of the room-temperature beamline has been demonstrated. The connection to the HD-EBIT gives access to few-electron ions up to the highest Z.

### 3.3. ARTEMIS

The ARTEMIS experiment uses a laser-microwave double-resonance spectroscopy scheme [38], which allows for precisely measuring the Zeeman substructure of the fine structure or hyperfine structure of the ion under consideration, which yields an ionic *g*-factor. When applied to the hyperfine structure, a combination of ionic *g*-factors can be used to determine the magnetic moments of the bound electron and of the nucleus. The principle of the laser-microwave double-resonance technique is to use fluorescence light from a closed optical transition as a probe for the microwave excitation between corresponding Zeeman sublevels. Different level schemes allow different measurement procedures, as has been detailed out in [38,39]. This spectroscopy scheme is conceptually different from the Stern–Gerlach-type experiments discussed above, in that it works with ions with non-zero spin nuclei, and also in that it works with ion ensembles rather than with single ions.

Figure 7 shows an overview of the experimental setup. The left-hand side is a schematic of the cryostat with the trap chamber, visible through the cutout of the surrounding superconducting magnet. It has a warm bore with 160 mm diameter and features a central field strength of 7 T at a field inhomogeneity of below 0.1 ppm in the trapping region.

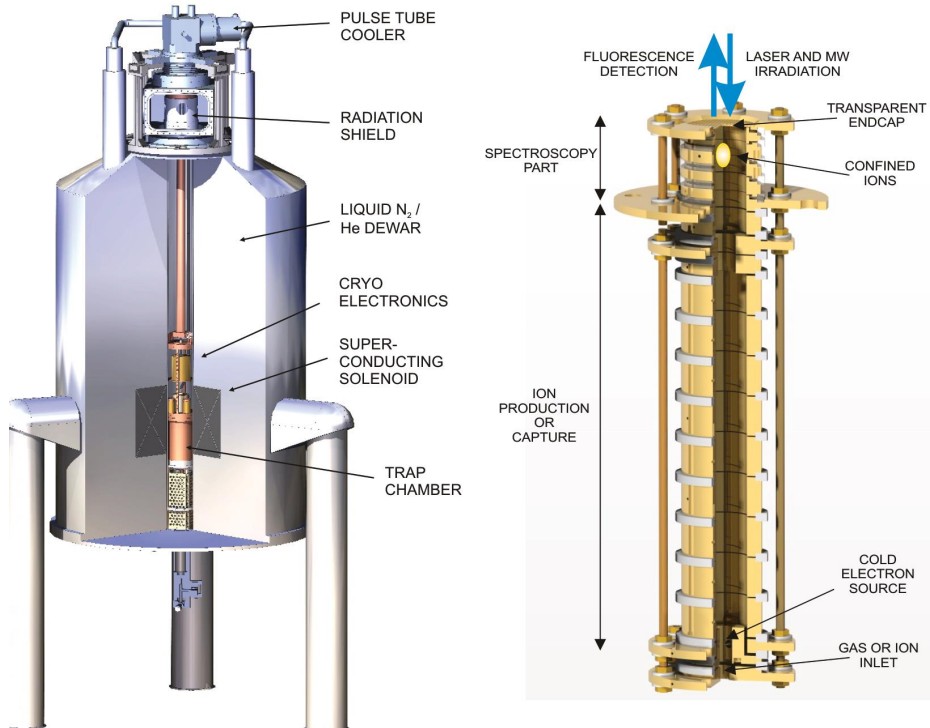

**Figure 7.** Schematic of the setup inside the superconducting magnet (**left**) and schematic of the Penning-trap arrangement (**right**).

The cryostat cooling the trap chamber and electronics is a pulse tube cooler inserted from the top. It cools the setup to 4 K with 1 W of cooling power. The trap chamber is acoustically isolated from the cold head by stranded wires to minimize vibration. The right-hand side shows a schematic of the Penning-trap arrangement, consisting of two main parts, one for spectroscopy (on the top) and the other for ion production or ion capture from external sources. The source part of the trap arrangement is basically a cylindrical open-endcap Penning-trap equipped with a cold electron source (HV field emission tip) and a cold gas source. The setup allows an electron beam to be reflected between the outer electrodes, ionizing present gas by electron-impact ionization similar to an EBIS in a reflectron mode [50]. The cold gas source is represented by a heatable baffle arrangement with initially frozen-out

gas that is released when heated. In this fashion, ions of certain charge state distributions can be produced like shown for argon in Figure 8 (left). Charge states can be selected by resonant ejection of unwanted species, and ions of interest are transported to the spectroscopy trap where they can be stored for periods of the order of days. The right-hand side of that figure shows the fine structure of $Ar^{13+}$ (boron-like argon) and the Zeeman effect due to the magnetic field of the trap. This ion species serves as a test system while commissioning of the setup. Also indicated in the figure are higher-order contributions to the Zeeman effect, which can be resolved by the envisaged combination of optical and microwave spectroscopy.

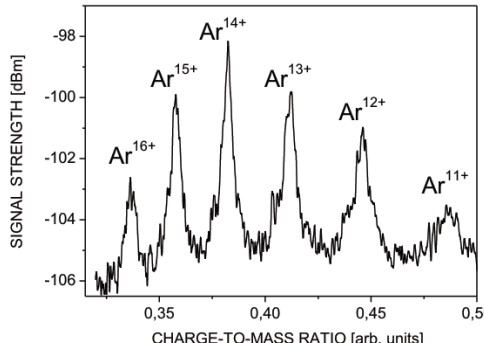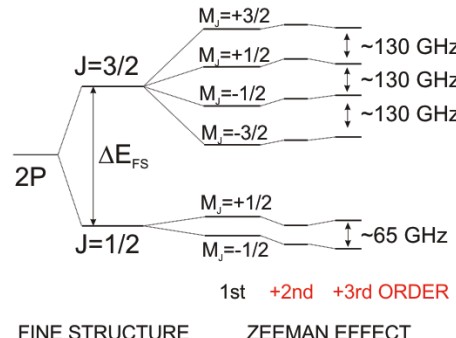

**Figure 8.** Charge state spectrum of in-trap created argon ions (**left**) and fine structure level scheme with Zeeman substructure of boron-like argon (**right**) with energy level shifts by second and third order contributions indicated (not to scale).

Quadratic, and partially also cubic, contributions to the Zeeman effect are within the design resolution, as has been detailed out in [39]. These allow laboratory access to individual higher-order contributions on the magnetic sector, in a regime not accessible before. The spectroscopy part of the Penning-trap arrangement is a dedicated development to the end of maximizing the optical fluorescence yield, a so-called "half-open" Penning-trap [52]. While the trap is open to the lower end in order for ions to be loaded, the upper endcap is optically open but electrically closed. This is achieved by use of a sapphire window with an ITO (Indium Tin Oxide) coating. The main advantage of the half-open design is the proximity of the trapping region to the window used for fluorescence collection, thus increasing the solid angle of detection by more than one order of magnitude with respect to conventional open-endcap designs. The ARTEMIS setup is currently being commissioned with in-trap created ions, and a connection to the HITRAP beamline in under way. Later, the experiment will be able to accept ions from the HITRAP facility, opening access to heavy, highly charged systems like $Pb^{81+}$ and $Bi^{82+}$, for which spectroscopic schemes exist [38].

## 4. Conclusions and Outlook

Following the successful test of strong-field QED with the Mainz *g*-factor experiment, next-generation setups are currently being developed within the HITRAP collaboration in order to push these tests towards the heaviest ions. However, the properties of these heavy systems give rise to a number of new challenges for the experiments. The exceptionally high ionization energies necessitate a connection to a suitable external ion source. While ALPHATRAP is connected to the Heidelberg EBIT, ARTEMIS will be supplied by the HITRAP facility. The experiments determine the *g*-factor from the measured ratio of the ions Larmor- to cyclotron frequency. While both experiments measure the trap eigenfrequencies using the image current detection method, they differ in the spin-state detection. ARTEMIS uses a laser- microwave double-resonance technique, which requires an accessible optical transition. In the regime of heavy ions, the hyperfine-structure transitions become optically accessible. Accordingly, ARTEMIS will address systems with non-zero nuclear spin, which

feature access to extremely strong magnetic fields. On the other hand, ALPHATRAP is based on the continuous Stern–Gerlach effect, which makes spin-state detection very challenging for heavy ions, but is applicable to all elements and charge states. Therefore, ALPHATRAP will determine the *g*-factor of nuclear spin-free systems with the highest precision. In combination, the two experiments will allow the most sensitive tests of the Standard Model in extreme conditions and allow the search for physics beyond the Standard Model.

**Author Contributions:** The authors have contributed equally to this work.

**Conflicts of Interest:** The authors declare no conflict of interest.

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
