# Peer review of "High-Precision Measurements of the Bound Electron’s Magnetic Moment"

_atoms, doi:10.3390/atoms5010004_

Reviewer 1 Report

Report on manuscript: atoms-165285

Title: Experiments for high-precision measurements of the bound electron's
magnetic moment Authors: Sven Sturm et al.

The authors summarize measurements of the magnetic moment (g-factor) of a bound electron performed so far and give a detailed account of the experimental setups, procedures and foreseen measurements. In these three Penning-trap experiments with highly charged ions they measured magnetic moments with ppb precision, serving as stringent tests of quantum electrodynamics 

calculations, and obtained access to properties of the ionic nuclei, to fundamental quantities like the fine structure constant and the atomic mass of the electron.

The paper is very well written, in particular the outlook towards an option to an independent determination of the fine structure constant and the g-factor of bound electron and nucleus in the ARTEMIS experiment makes this paper worthwhile to publish.

I have just a few specific remarks:

The title sound a bit like a round wheel, I propose: High-precision measurements of the bound electron's magnetic moment

In the abstract: Highly charged ions represent environments which allow to subject one or more bound electrons to unsurpassed electromagnetic fields.

Even in a neutral atom are the 1s electrons subjected to unsurpassed electromagnetic fields.

But: Highly charged ions represent environments which allow to study precisely one or more bound electrons subjected to unsurpassed electromagnetic fields.

Page 2: … setup that is used to perform…

For medium-Z ions, a conventional EBIS (Electron Beam Ion Source) ref. to paper by Donets et al. may be employed.

To my opinion the EBIT is used here as an ion source and is thus an EBIS and should be called like that.

Page 3: Also EBIT's are used as sources by extracting the ions from them after breeding (ref. D. Schneider et al. Phys. Rev. A42 (1990) 3889). Compact EBIT/S using permanent magnets and electron energies up to 15 keV (’Dresden EBIT’) are commercially available.

… remaining parts, which has to be actuated fast in order to prevent too…..

…….trap volume has been demonstrated (ref. needed!). Consequently,……

Page 7: …a test of the description of the inter-electronic interaction between the 3 electrons in the…..

Page 9 …..continuous Stern-Gerlach effect, as well as, the…….

Page 10 Fig.6   remove (J. Crespo) and insert HCI from tt-EBIT

Author Response

Dear Editor, dear Reviewer, 

We would like to sincerely thank the reviewer for the comments and valuable suggestions to the manuscript. We have implemented all suggestions into the attached revised version. 

Best regards,

SVEN STURM, on behalf of the authors.

Reviewer 2 Report

The Authors review the current status of the high precision Penning-trap experiments measuring the magnetic moment of highly charged ions i.e. "Mainz g-factor experiment",
ALPHATRAP at MPI-K Heidelberg and ARTEMIS  GSI, Darmstadt. Undoubtedly, the research is carried at the top scientific level. These measurements provide the high quality data, with ppb precision and better, that juxtaposed with the most advanced QED calculations
can be used for mutual verification of theory and experiments as well as, very important applications like detmination of physical constants (e.g the atomic mass of the electron), probing the nuclear structure properties (e.g. nuclear charge radii from the isotope shifts) and
searches for physics beyond the Standard Model.

Definitely, it can be considered as review manuscript to be published in Atoms. It provides concise updates on the latest progress made in the research of highly charged ions and g-factor measurements.

However, the article can be improved/completed to be more precise in several places.

1. In the abstract and in the text, there are comments on an alternative determination
of the fine structure constant in Ref.[31,32]. It looks very important to the Authors,
and indeed is actually justified. I would encourage them to provide some precise informations e.g. what is a necessary experimental precision to improve the CODATA value.

2. The text seems to be a compilation of fragments written by different authors. In some places it is too visible, and can be smoothed a little. For example:

In section "2. Sources ... " Parts on lines 66-78 "Low- and medium-Z ions" and lines 79-88 "For medium-Z ions".

- (Electron Beam Ion Trap) from 79 move to line 66 to introduce "EBIT"
- I guess that the first part is related to "Mainz g-factor experiment", but it is not clear from the text.  Please, clarifify this. Otherwise, some informations look in contradiction e.g. lines 78-79 vs. line 82.

3. Table 1. Please, align the numbers to dots

4. line 164, I suggest replace :  "validity of the g-factor calculation" --> "validity of the g-factor theory"

5. lines 238-240 : Comment on : "the theoretical prediction is not yet sufficient ...".  The theoretical prediction for high-Z is limited due to unknown the nuclear structure effects. In practice, it is impossible to match the accuracy of the present \alpha for high Z charges.

Author Response

Dear Editor, dear Reviewer, 

We would like to sincerely thank the reviewer for the comments and valuable suggestions to the manuscript. We have implemented all suggestions into the attached revised version. 

Particularly, we have clarified the specific subject of the first part of chapter 2 by adding a appropriate sentence. Furthermore, we have added additional information of the prospects and goals of the determination of alpha from light and heavy HCI.

We hope that the revised manuscript is suitable for publication.

Best regards,

SVEN STURM, on behalf of the authors.

Reviewer 3 Report

The authors present a review of the current status of experiments for high-precision measurements of the bound-electron g factor. It includes the successful measurements which have been performed in the framework of the Mainz g-factor experiments as well as the very recent developments within the ALPHATRAP and ARTEMIS projects. These investigations have already provided the most stringent tests of bound-state QED calculations
of middle-Z ions and the most precise determination of the electron mass. It is expected that the extension of these measurements to higher Z ions and to ions with nonzero nuclear spin, that is anticipated in the framework of the ALPHATRAP and ARTEMIS projects,  will give new opportunities for tests of the QED calculations at strong coupling regime and precise determinations of the fine structure constant and the nuclear magnetic moments. I strongly recommend this paper for publication in Atoms. The paper is well written and can be published as it is.

Author Response

Dear Reviewer,

we would like to thank the reviewer for the positive evaluation of our manuscript.

Best regards,

SVEN STURM on behalf of the authors.